# Legal Financial Obligations and Probation: Findings from the 1995 Survey of Adults on Probation

**Marshall L. White \* and William J. Sabol**

Department of Criminal Justice and Criminology, Georgia State University, Atlanta, GA 30303, USA;
wsabol@gsu.edu
\*  Correspondence: mwhite129@student.gsu.edu

**Abstract:** During the late 20th century, imprisonment rates in the United States saw unprecedented growth, leading correctional systems across the country to face widespread overcrowding and underfunding. Subsequently, policy makers sought out alternatives to incarceration for certain categories of offenses. Community supervision, such as probation, emerged as a popular solution to both reduce prison and jail populations as well as to generate revenue to fund the rapidly expanding legal system. With the rise in community supervision came increases in the number of people on probation for lower-level and non-violent offenses. The expansion of legal financial obligations (LFO) placed additional burdens on these persons, who disproportionately sit in lower socio-economic status brackets. Using data from the 1995 Survey of Adults on Probation (SAP), the current study adds to the literature on probation and LFOs in an important way. The SAP data contain information on the amount, frequency, and type of LFO. Thus, this paper examines the distinct types of LFOs to determine the differential burden that each type of LFO has on people on probation. This paper finds that of all types of fees, those associated with victim restitution are most likely to lead to missed payments, while those that generate revenues do not contribute significantly to missed payments. This paper discusses the implications of this for procedural justice and fairness.

**Keywords:** probation; legal financial obligations; monetary sanctions; criminal justice

## 1. Introduction

Mass incarceration has been at the forefront of criminal justice research and policy discussions for decades, but reforms stemming from mass incarceration have only just begun to share the spotlight in this critical space. During the current era of reform, community supervision, such as probation or parole, emerged as one of the leading solutions to manage people convicted of lower-level offenses while avoiding the more costly incarceration term. While ostensibly an alternative to incarceration, probation populations grew along with prison populations, expanding the net of persons under the authority of correctional authorities. It was not so much that those who would have formerly served their sentences in prison or jail are now serving their sentences in the community. Rather, persons who might not have been placed on community supervision in the first place were ordered to serve sentences in the community. Consequently, some have called the phenomenon "mass probation" (Phelps 2020). Indeed, recent data from the United States Department of Justice, Office of Justice Programs, Bureau of Justice Statistics (2019) estimate that 4.3 million adults are currently on probation compared to 1.4 million who are imprisoned. It is therefore important to understand how people experience this form of correctional control and to question whether mass incarceration reforms are having their anticipated effect.

Scholars have pointed out that although probation is intended to be an alternative to incarceration, it often just delays an inevitable stint in custody due to the requirements associated with probation (Klingele 2013; Phelps 2020). Legal financial obligations (LFOs) have emerged as an especially arduous requirement for some people on probation, highlighting

inequalities based on wealth in our legal system. As Ruhland (2021, p. 69) writes, "Since payment is a condition of probation, if one is not paying them [legal financial obligations] the individual might receive a sanction or revocation." Thus, community supervision becomes a delayed feeder into custody. Legal financial obligations can additionally extend the original probation sentence if payments are not made, keeping people experiencing poverty in the system for a longer period. Importantly, we interrogate the use of LFOs in our legal system, especially as they pertain to probation, where LFOs are ubiquitous.

In this paper, we tap the 1995 Survey of Adults on Probation to ask, what types of legal financial obligations are associated with missing a payment toward monetary sanctions? By answering this question, we aim to add to the literature on the philosophical purposes of legal financial obligations as a form of punishment. Namely, we argue that given our findings and the prior literature on LFOs, more attention should be paid to how their philosophical purpose, or lack thereof, impacts the perceived legitimacy of community supervision. Understanding the disparity in the philosophical intent of each type of LFO versus the actual practice of assessment and collection of them may give us insight into the legitimacy issue our criminal justice system is facing.

## 2. Literature Review

### 2.1. Legal Financial Obligations in the United States

The United States has a long history involving the use of monetary sanctions. On the legal side, the Eighth Amendment to the United States constitution, ratified in 1791, explicitly outlaws excessive bail and fines (U.S. Const. amend. VIII). Debtor's prison was later outlawed in 1833 (Hager 2015). Contemporary legal challenges to LFOs include *Williams v. Illinois* (1970) and *Bearden v. Georgia* (1983). *Williams* established that incarceration for non-payment of LFOs cannot exceed the maximum punishment for the offense. *Bearden* found that no person who is declared indigent by the court shall be incarcerated for non-payment of LFOs. More recently, states have begun enacting policies to curb the collateral consequences of legal debt. For example, it is now illegal in many states to suspend driver's licenses for failure to make payments toward legal debt. As of 2021, 13 states have these policies in place and 11 additional states have policies in the works (American Civil Liberties Union 2021). Regardless of the legal challenges, LFOs remain a common form of punishment, and there is evidence to support the claim that their use has grown exponentially over the past few decades (Greenberg et al. 2015; Harris et al. 2010). Accordingly, it is important to distinguish between the types of legal financial obligations one can accrue through contact with the legal system.

Legal financial obligations, also known as monetary sanctions, include but are not limited to, fines, fees, restitution, court costs, surcharges, and interest. Bail is an additional monetary sanction. However, it is outside the realm of this paper and is not included in our analyses. While they are often grouped together under the umbrella of monetary sanctions, LFOs vary in their philosophical purpose, in their assessment, and in their collection. Here, we briefly discuss the definition of each type of LFO that is included in our study. The purpose of this exercise is to highlight their differences before engaging in a discussion on their philosophical purpose.

Fines are monetary penalties assessed by judges upon conviction of an offense for a certain crime (Ruback et al. 2014). They are punitive in nature. Judges may look to previous statutes when assessing fines, but in certain jurisdictions, such as municipal courts, judges have a great deal of discretion when determining how much to charge (Martin et al. 2018). Notably, fine amounts may vary across and within states.

While fines are intended to be a punitive measure, the purpose of court costs and fees is to recoup the costs of criminal justice system operations. Also known as "user fees," court fees may include clerk's fees, docket fees, witness summons' fees, district attorney fees and a fee for the use of a public defender. Some court fees are mandated by statutes, but others are left to the judge's discretion. Other types of fees include supervision fees, which are typically assessed by a probation agency for the use of their services. Supervision fees are

especially relevant to people convicted of lower-level offenses as community supervision is one of the most common sentences for these crimes. Other fees that can be incurred through supervision include electronic monitoring fees, drug testing fees, and a baseline fee for participation in programs for drug treatment (Greenberg et al. 2015).

Restitution is a court-ordered payment made by an offender to a crime victim and is intended to compensate for damages caused by the crime (Ruback et al. 2014). Unlike fines and fees, the goal of restitution is the pursual of restoration and accountability to crime victims (Martin et al. 2018). Restitution may represent an anomaly in discussions about the financial impacts of monetary sanctions on people found guilty of a crime. This is due in part to the potential rehabilitative effects that restitution has been shown to have for both offenders and victims (Ruback et al. 2018; Cares and Haynes 2018).

All other monetary sanctions such as surcharges, interest, and fees fall into a similar category as court costs. These monetary sanctions are not assessed by judges as a part of an individual's sentence. Rather, these are "collateral" monetary sanctions that come with participation in the legal system. For example, a judge may sentence a person to a $500 fine and a 2-year probation term. However, the true financial cost of the sentence will likely be much greater if the guilty party has to pay for things such as supervision fees, drug testing fees, or electronic monitoring. These costs and fees all go toward operating the justice system and are not intended to serve as punishment. However, if a person on probation falls behind on any of these payments, their probation may be prolonged or revoked. Thus, it is important to include them in studies of legal financial obligations because of their punitive ability.

Because court systems in the United States are fragmented (e.g., municipal, state, and federal), and having varying levels of discretion when meting out punishment, it is difficult to obtain a nationally representative estimate of average LFO debt for people in the legal system. However, some scholars have attempted estimations using court records in combination with other data. Greenberg et al. (2015) analyzed two hundred thousand court records in Alabama that contained information on monetary sanctions. Based on their analyses, they estimated that outstanding debt attributed to LFOs rose from 260 million in 2005 to 13 billion in 2013. It was additionally estimated that defendants were assessed a median of $2000 in monetary sanctions in 2005, up from $1000 in 1995. Harris et al. (2010) found that in Washington state in 2004, defendants were assessed a median of $1347 in monetary sanctions, and the lifetime accrual of LFO debt amounted to $7234. The Alabama Appleseed Center for Law and Justice's (2018) survey of adults with legal debt found that respondents reported owing a median amount of $2700.

Criminal legal debt is unique because unlike other types of debt, it cannot be taken care of through conventional methods such as loan forgiveness or bankruptcy proceedings. Moreover, you can be jailed for non-payment of legal debt, but you cannot be jailed for other types of debt (e.g., medical and student). Indeed, although debtor's prison was outlawed in the early 19th century, the practice of jailing people for willful non-payment of LFOs persists, leading some to call it the modern debtor's prison (Sobol 2015; Colgan 2018). Willful non-payment means the judge found that you are not paying toward your legal debt when you have the means to. Often, these decisions are not made without taking into consideration the defendant's full financial history, resulting in incarceration for those who are financially unstable. Incarceration is not the sole consequence of non-payment of LFOs. Scholars have documented a range of collateral consequences of non-payment, including but not limited to, extended probation sentences, driver's license suspension, and disenfranchisement (Martin et al. 2018).

Given the disparate legal outcomes caused by monetary sanctions, it is critical that researchers begin to interrogate how the philosophical purpose of their use impacts the legitimacy of the legal system. In the next section, we highlight popular philosophical and political justifications that have been used to defend the assessment of LFOs as a form of punishment.

### 2.2. Philosophical Purposes of Legal Financial Obligations

Equal justice under law is the basic premise of our legal system in the United States. This doctrine means that United States citizens should be held equally accountable under the law regardless of extraneous factors such as wealth. Yet, the use of LFOs as a form of punishment in the United States enforces a two-tiered justice system based on wealth whereby monetary sanctions are a temporary inconvenience for wealthy defendants, but for people experiencing poverty, they can have lifelong effects. In criminal law it is generally agreed that punishment should serve one of five philosophical purposes: retribution, rehabilitation, deterrence, incapacitation, or restoration. Lawmakers frequently cite retribution and deterrence as justification for some monetary sanctions (fines) and restoration and rehabilitation for others (restitution) (Harris 2016; Ruback 2014). However, mounting evidence shows that there have been substantial difficulties in translating the philosophical purposes of LFOs as punishment into actual practice.

First, as mentioned, there is the issue of disparities in punishment based on wealth. Ruback (2014) asks, if monetary sanctions are designed to punish and deter, what impact do they have on the wealthy? For example, a $500 fine assessed to someone earning an income below the poverty threshold will have a more punitive effect than the same fine assessed to someone earning at or above the median household income. People on probation may have their probation extended or revoked if they become delinquent on payments. Thus, critics have questioned whether lawmakers and court officials can claim deterrent and retributive purposes for fines when they have dramatically different impacts on people convicted of the same offense (Harris 2016).

Secondly, court costs and other fees are used to generate revenue for state and local funds. The use of LFOs to fund the legal system creates a conflict of interest whereby law enforcement and legal actors are incentivized to arrest and prosecute people. In 2014, the Department of Justice, Civil Rights Division investigated the Ferguson Police Department for constitutional rights violations related to this practice. They found that, "the City budgets for sizeable increases in municipal fines and fees each year, exhorts police and court staff to deliver those revenue increases, and closely monitors whether those increases are achieved" (United States Department of Justice, Civil Rights Division 2015). As a result, some scholars have begun studying whether monetary sanctions impact the legitimacy of institutions such as the courts (Pleggenkuhle et al. 2021).

Finally, while restitution is perceived by some as the most defensible LFO based on its restorative and rehabilitative effects (Ruback et al. 2018), there is incongruence between this perception and the actual implementation. In lower-level felonies and misdemeanors, restitution is often not paid directly from an offender to a victim. Rather, the offender makes payments to a Victim's Assistance Fund which then pays out the victim. Thus, the restorative and rehabilitative functions of restitution are lost during the process of indirect restitution (Martin and Fowle 2020). At the federal level, the United States Government Accountability Office (2020, p. 1) stated that the goal of restitution is to "restore victims of federal crimes to the position they occupied before the crime was committed." Yet, in most federal cases from 2014 to 2016, restitution was ordered for victimless immigration or drug trafficking offenses. During this time, judges ordered 33.9 billion in restitution but only 2.95 billion was collected (United States Government Accountability Office 2018). As of 2016, outstanding restitution debt amounted to 110 billion and 100 billion of that is due to an inability to pay.

Indeed, uncollected restitution seems to be a longstanding issue. The Office for Victims of Crime previously supported a project aiming to highlight promising practices for victims' services such as restitution. The report cites the prioritization of revenue generating LFOs as an obstacle that directly influences the success of restitution programs (United States Department of Justice, Office of Justice Programs, Office for Victims of Crime 1998). So, in addition to restitution being commonly assessed in victimless crimes, there are also issues with collection and disbursement. Fines, fees, and restitution are intended to serve multiple philosophical purposes of punishment (e.g., deterrence, retribution, and restoration). The

actual practice of assessing and collecting LFOs, however, illustrates a system that is at odds with these purposes.

*2.3. Current Study*

We use nationally representative data collected by the Bureau of Justice Statistics (BJS) to parse out different types of legal financial obligations and the likelihood of missing a payment. The BJS defines probation as "a court-ordered period of correctional supervision in the community, generally as an alternative to incarceration. In some cases, it may be a combined sentence involving incarceration followed by a period of community supervision." Although the probation population has steadily decreased since 2008, 1 in 59 people in the United States remain under this form of correctional control (United States Department of Justice, Office of Justice Programs, Bureau of Justice Statistics 2019). The 1995 Survey of Adults on Probation allows us to examine the amounts of regular, monthly payments and payment types of a sample of probationers. We use these variables to determine what types of payments, among other factors, are associated with missing a payment. We hypothesize:

**Hypothesis 1 (H1).** *As the number of fees/fines increases, the chance of missing a payment will increase.*

**Hypothesis 2 (H2).** *Fees associated with criminal justice operations (e.g., probation supervision, treatment costs, and court costs) will be less likely to be missed than those associated with restorative justice, such as victim restitution.*

**Hypothesis 3 (H3).** *Persons with lower incomes and who receive public assistance will be more likely to miss payments than those with higher incomes.*

Our hypotheses are based on prior literature on fines and fees which indicates that economically marginalized populations are at a higher risk of accruing legal debt (Martin et al. 2018; Alabama Appleseed Center for Law and Justice 2018). We hypothesize that revenue generating LFOs will be less likely to be missed because the monies generated by these LFOs are necessary for continued justice system operations. Additionally, probationers may perceive these LFOs to have more punitive consequences attached if they miss a payment (e.g., technical violations and revocation).

**3. Methodology**

*3.1. Data Source*

Data for the current paper were located using the Inter-University Consortium for Political and Social Research (ICPSR). In 1995, the Bureau of Justice Statistics conducted a one-time Survey of Adults on Probation (SAP) (United States Department of Justice, Office of Justice Programs, Bureau of Justice Statistics 2006). The SAP is unique because it includes comprehensive measures of legal financial obligations, something that most publicly available datasets do not have. Specifically, it asks about regular and lump sum payments toward: probation fees, fines, court fees, drug test fees, restitution, community service fees, and treatment fees. Although the data are from 1995 and can be construed as somewhat dated, we argue that since the BJS conducted the SAP, the use of fees has expanded; therefore, results from the SAP may underestimate the effects of fees on payments. Further, as the BJS included measures for LFOs in a 1995 survey, and data show that their use has expanded over time, it proves that this is a longstanding issue (Harris et al. 2010). Moreover, to our knowledge, the 1995 SAP is the only large-scale effort that attempts to measure legal financial obligations at the national level, something that has repeatedly been called for in recent literature on LFOs.

Other studies using the 1995 SAP data are primarily descriptive and include those from BJS statisticians (e.g., Bonczar 1997; Harlow 2003; Maruschak 1999; Mumola 2000) as well as from analysts in other federal agencies (Thompson 2011). The SAP data have

also been used to study homelessness and mental health problems among correctional (Katz 2003), and to study non-compliant behavior among probationers (Schulenberg 2007). In her study of non-compliant behavior, Schulenberg focused on differences by the sex of persons on probation in several measures of non-compliance, including missing a regular fee payment. She examines how the sex of the person on probation interacts with race and measures of social disadvantage to conclude that females on probation experience multiple inequalities of race, gender, and socio-economic status that contribute to non-compliant behaviors such as missing payments. Our work focuses on missed payments but focuses on the total number of and types of fees imposed to assess whether any of these affect the probability of missing a recurring payment.

### 3.2. Sample and Data Collection Procedures

The SAP has two datasets, one based on an administrative record-checks collection effort and the other based on personal interviews. The sample designs for both were similar with one modification for the personal interview sample, as described below. The sample for the 1995 SAP was selected from a universe of 2637 state, county, and municipal probation agencies responsible for supervising a total of 2,618,132 persons formally sentenced to probation. The universe included adults convicted of felonies or of misdemeanors. The sample was a stratified, two-stage design. The first stage consisted of probation agencies stratified by government branch (executive or judicial), level (state or local), and census region (Northeast, Midwest, South, or West). After selecting the largest agencies to be self-representing (i.e., certainty agencies), randomly selecting one agency from the remaining strata, and selecting two additional subagencies, the number of agencies in the SAP records-check sample was 167. In the second stage, a systematic sample of approximately 1 in 442 persons on active probation were selected for the records-check sample.

From the records-check sample, 4703 persons on probation were selected for personal interviews. Persons who were not on active probation were excluded from the personal interview sample; hence, the universe for the personal interview sample was 2,065,896. Additionally, as with the records-check sample, excluded from the personal interview sample were persons under 18 years of age, those who were not formally sentenced to probation, those who had absconded, those who were supervised by a Federal agency, those who were only under parole supervision, and those under pretrial diversion only.

For the personal interview sample, 122 agencies were selected. 21 agencies declined to participate in the personal interviews. After excluding agencies that would only participate in the records-check but not the personal interview sample, the final sample was 101 probation offices in which interviews were conducted. U.S. Census Bureau field staff conducted the personal interviews. A total of 4703 persons were sampled for a personal interview. The interview process included asking probation office personnel to make initial contact with a sampled person on probation to describe the SAP and to schedule a personal interview to coincide with a regular office visit when possible, and to follow-up and encourage sampled persons to participate. All interviews were conducted in the probation office. The SAP documentation does not describe whether the spaces where the interviews were conducted were separate from where office personnel generally conduct their business. Based on our understanding of U.S. Census Bureau interview procedures, we believe that the Census field staff arranged for interviews to be conducted in a location that provided the interviewed persons with privacy. Interviews were conducted face to face. A 50% response rate was achieved.

For both surveys BJS developed survey weights to adjust base weights; for the interview sample, the adjustments to base weights (probabilities of selection) included site non-interview (i.e., to address exclusion of sampled sites that elected not to participate), and person non-interview (i.e., to adjust for sampled persons for whom an interview was not obtained).

The sample was designed to generate national- and not-state-level estimates. Hence, state-specific issues cannot be addressed with the SAP data. The sample data are a good fit

for our theoretically defined population. However, omitting absconders from the sample could present sample selection bias. People abscond for many reasons, but one potential reason could be that they are not able to pay the fee associated with a visit to a probation office. Excluding people on pretrial diversion or supervision will also eliminate a portion of the population who are also paying fees. We are not aware of any pretrial diversion program that is truly free to participate in. Even if there is not a fee associated with the program, participants are tasked with other financial obligations such as securing transportation to probation meetings, taking off work, and finding childcare. The same is true of both active probationers and absconders. All of that said, sample selection bias may exist in the analysis because we will only be able to speak on the experiences of active probationers, who may already have the financial means to successfully participate in their probation.

### 3.3. Variables

Dependent Variable: The key dependent variable in our analysis is *missed pay*, which is a dichotomous variable (1 = yes 0 = no) measured by the question, "Did you miss any required payments during the last 12 months?"

Independent Variables: Our key independent variables are:

- Number of regular fees/fines imposed. This is a count of the total number of types of fees/fines imposed in which regular payments were required, where the count is of the number of different types of fees/fines. The mean number of fee/fine types imposed was slightly more than 2 (Table 1) and the number per person ranged from 0 to 8. We derived this measure from two SAP survey items that asked, "Were you required to make any REGULAR payments during the last 12 months as a result of your conviction on [DATE}?" and "Were you required to make ANY OTHER REGULAR payments?" The response options to these questions included:

  - Probation supervision fees,
  - Fines,
  - Court costs/public defender fee,
  - Drug testing/other lab fees,
  - Monetary restitution,
  - Community service fee,
  - Counseling/treatment fee, and
  - Other (specify).

**Table 1.** Summary statistics on variables used in the analysis for the sample of persons on probation who were ordered to make at least one type of regular fee/fine payment.

| Variable | | N. Obs. | Mean | Std. Dev. | Minimum | Maximum |
|---|---|---|---|---|---|---|
| Missed a regular payment | | 1557 | 0.42 | 0.49 | 0 | 1 |
| Number of different types of regular fees/fines | | 1557 | 2.01 | 1.25 | 0 | 8 |
| Fee/fine types and amounts | | | | | | |
| | Restitution ordered | 1557 | 0.24 | 0.43 | 0 | 1 |
| | Court costs ordered | 1557 | 0.40 | 0.49 | 0 | 1 |
| | Probation fees ordered | 1557 | 0.73 | 0.45 | 0 | 1 |
| | Fines ordered | 1557 | 0.42 | 0.49 | 0 | 1 |
| | Drug treatment fees ordered | 1557 | 0.09 | 0.28 | 0 | 1 |
| | Community service fees | 1557 | 0.05 | 0.22 | 0 | 1 |
| | Counseling/treatment fees | 1557 | 0.06 | 0.23 | 0 | 1 |
| | Other regular fees | 1557 | 0.03 | 0.18 | 0 | 1 |
| | Lump sum payment ordered | 1557 | 0.28 | 0.45 | 0 | 1 |
| Monthly payment amount * | | 1452 | 65.07 | 74.37 | 2 | 960 |
| Offense type: | | | | | | |
| | Murder/homicide | 1867 | 0.01 | 0.11 | 0 | 1 |
| | Rape/sexual assault | 1414 | 0.05 | 0.21 | 0 | 1 |
| | Robbery | 1414 | 0.02 | 0.13 | 0 | 1 |
| | Aggravated assault | 1414 | 0.08 | 0.28 | 0 | 1 |

**Table 1.** *Cont.*

| Variable | | N. Obs. | Mean | Std. Dev. | Minimum | Maximum |
|---|---|---|---|---|---|---|
| | Simple assault | 1414 | 0.01 | 0.11 | 0 | 1 |
| | Other violent | 1414 | 0.00 | 0.07 | 0 | 1 |
| | Property | 1414 | 0.35 | 0.48 | 0 | 1 |
| | Drug | 1414 | 0.21 | 0.40 | 0 | 1 |
| | Public order | 1414 | 0.27 | 0.44 | 0 | 1 |
| Offense severity: Felony offense | | 1557 | 0.65 | 0.48 | 0 | 1 |
| Time on probation (months) | | 1220 | 19.74 | 20.06 | 0 | 192 |
| Past-year total personal income categories, in SAP categories and 1995 dollars * | | | | | | |
| | Under $1000 | 1491 | 0.07 | 0.25 | 0 | 1 |
| | 1000–1999 | 1491 | 0.05 | 0.22 | 0 | 1 |
| | 2000–2999 | 1491 | 0.05 | 0.21 | 0 | 1 |
| | 3000–3999 | 1491 | 0.05 | 0.21 | 0 | 1 |
| | 4000–4999 | 1491 | 0.04 | 0.19 | 0 | 1 |
| | 5000–5999 | 1491 | 0.06 | 0.24 | 0 | 1 |
| | 6000–7499 | 1491 | 0.06 | 0.24 | 0 | 1 |
| | 7500–9999 | 1491 | 0.09 | 0.28 | 0 | 1 |
| | 10,000–11,999 | 1491 | 0.08 | 0.28 | 0 | 1 |
| | 12,000–14,999 | 1491 | 0.12 | 0.32 | 0 | 1 |
| | 15,000–19,999 | 1491 | 0.11 | 0.31 | 0 | 1 |
| | 20,000–24,999 | 1491 | 0.08 | 0.28 | 0 | 1 |
| | 25,000–49,999 | 1491 | 0.11 | 0.31 | 0 | 1 |
| | 50,000 or greater | 1491 | 0.03 | 0.16 | 0 | 1 |
| Received welfare payments | | 1557 | 0.15 | 0.36 | 0 | 1 |
| Age in years | | 1557 | 32.62 | 10.60 | 16 | 77 |
| Race | | | | | | |
| | Person was White | 1557 | 0.67 | 0.47 | 0 | 1 |
| | Person was Black | 1557 | 0.27 | 0.45 | 0 | 1 |
| | Person was of another race | 1557 | 0.05 | 0.22 | 0 | 1 |
| Hispanic: Person was Hispanic | | 1543 | 0.14 | 0.34 | 0 | 1 |
| Sex: Person was Male | | 1557 | 0.77 | 0.42 | 0 | 1 |
| Education levels | | | | | | |
| | Up through 8th grade | 1531 | 0.07 | 0.25 | 0 | 1 |
| | 9th through 11th grade | 1531 | 0.21 | 0.40 | 0 | 1 |
| | High-school graduate | 1531 | 0.33 | 0.47 | 0 | 1 |
| | GED ** | 1531 | 0.11 | 0.32 | 0 | 1 |
| | College/graduate school | 1531 | 0.29 | 0.45 | 0 | 1 |
| Owned home | | 1518 | 0.35 | 0.48 | 0 | 1 |
| Rented | | 1518 | 0.57 | 0.49 | 0 | 1 |
| Contributed to mortgage/rent | | 1456 | 0.77 | 0.42 | 0 | 1 |
| Region | | | | | | |
| | Northeast | 1557 | 0.11 | 0.31 | 0 | 1 |
| | Midwest | 1557 | 0.14 | 0.35 | 0 | 1 |
| | South | 1557 | 0.62 | 0.48 | 0 | 1 |
| | West | 1557 | 0.13 | 0.33 | 0 | 1 |

Source: Authors' analysis of SAP data. Note: * Payment amounts and income categories are shown in SAP amounts and levels. ** Completed Test of General Educational Development, equivalent to a high school diploma.

The text responses to the "Other (specify)" were not included in the archived version of the SAP. A total of 66 persons responded that they were required to pay "Other" types of fees. Examples of other types of fees/fines not explicitly identified may include pretrial services fees, health services, or mental health services fees.

Dummy (or indicator) variables for each type of fee/fine; each dummy variable equals 1 if a specific type of fee was imposed and zero otherwise. The fee types include those listed above in the description of the number of regular fees/fines imposed. Just under one-quarter (24%) of the sample were ordered to pay restitution; 40% were ordered to pay court costs; another 43% were fined; and 73% to pay probation fees. The proportion ordered

to pay probation fees was highest among all fee/fine types. For the lesser used categories of fees, approximately 9% were ordered to pay drug treatment, while approximately 5% were ordered to pay community service or counseling fees, and 3% to pay other regular fees. More than one-quarter (27%) were ordered to pay a lump-sum amount (Table 1).

- Monthly payment amount: This is the monthly amount of regular payments due, summed across all types of fees. The mean monthly amount was $65 in 1995 dollars (or $117 in current dollars.) We created the monthly payment amount from two variables: The first is a SAP question that asks for the amount of regular payments, and the second is from the frequency with which the regular payments are to be made. For the frequency of regular payments, three responses are available: Week, Month, and Other (specify). If weekly payments were required, we multiplied the payment amount by four (4) weeks per month (on average) to get a monthly amount. If monthly payments were required, we used the payment amount as the monthly amount. For the open-ended "Other (specify)," as occurred with the variable for other types of payments, the SAP data did not include the text descriptions for the other frequencies of payments. To estimate the frequency of payments when "other" was specified as the periodicity, we did the following: For persons on probation who had both regular and other regular payments, we used the week or monthly frequency in the observed payment category as an estimate of the interval for the unobserved interval, and for the remainder, we set the interval equal to semi-annual payments and divided the amount by six (6) months per semi-annual year. This resulted in monthly amounts that were comparable to those reporting that they were required to make monthly payments.

- Offense category: The SAP provides detailed offense codes for the "controlling offense," which is defined as the offense with the longest term. We classified the detailed offense codes into several offense categories that include

    - Murder/homicide,
    - Rape/sexual assault,
    - Robbery,
    - Aggravated assault,
    - Simple assault,
    - Other violent,
    - Property,
    - Drug, and
    - Public order.

We coded each offense category as a dummy variable equal to 1 if the named offense was the controlling offense, and zero otherwise. We used murder/homicide as the excluded dummy variable in regressions. Collectively, the violent offenses—murder, rape, robbery, aggravated assault, simple assault accounted for approximately 17% of the sample. Property offenses were the most commonly occurring offenses, as 35% of persons on probation were held for a property offense. Drug offenses amounted to 21% of the sample, and public order offenses another 27%. The distribution of offenses of persons on probation derived from the personal interview sample aligns with the distribution obtained from the record-checks SAP sample, as reported by Bonczar 1997: Table 1). Offense severity level was measured as a dummy variable equal to 1 if the person on probation was sentenced for a felony offense, and zero otherwise. Almost two-thirds (65%) of the sample was held for a felony offense.

- Time on probation in months: This measures the duration of time between a probationer's entering supervision and their SAP interview date. The mean time on probation until the SAP survey date was approximately 20 months. Because the SAP are of a cross-section of actively-supervised probationers, those still active on the survey date are more likely to have been on probation for a longer period of time than the average probationer. As the active probationer sample may be more likely to meet

probation requirements, we wanted to capture effects for persons who have spent less time on probation. To that end, we introduced time on probation into our regressions as both linear and non-linear (squared) terms.

- Prior year personal income: The SAP asked respondents to classify their past-year personal income into one of 14 income categories; the lowest category was no income and the highest category was $50,000 or more. We re-classified these 14 categories into 7 categories that included less than $5000; $5000 to less than $10,000; 10,000 to less than 15,000; 15,000 to less than 20,000; 20,000 to less than 25,000; 25 to less than 50,000; and 50,000 plus. We hypothesize that income affects ability to pay fees, and consequently whether a person misses a payment. In our analyses, we used income in 1995 dollars, but for presentation purposes, we used inflation-adjusted category labels that show the 1995 income levels in current (2021) dollars. Almost half (46%) of the sample reported past-year personal income of less than $10,000 in 1995 dollars (or $18,000 in current dollars); 39% reported income between $10,000 and $25,000 in 1995 dollars (or $18,000 to $45,000 in current dollars); and 14% reported past-year income of over $45,000.
- Welfare recipiency. We measured this using a dummy equal to 1 if a probationer received welfare during the year prior to their interview. Fifteen percent of the sample reported receiving welfare payments during the past year.

Control Variables included age, race, Hispanic origin, sex, education, whether a person owns or rents a house or apartment, whether a person contributes to the mortgage or rent, and Census region. We measured these variables as follows:

- Age was measured by age in years. The mean age of the sample was 33 years.
- Race: Race was measured by three dummy variables, one if a person reported being White, another if a person reported being Black, and a third if a person reported either Asian/Pacific Islander or Native American Alaska Native. We recognize that grouping Asian/Pacific Islanders and Native Americans makes the category heterogeneous; however, the numbers of persons in these two categories were small and whether we used the four category race classification in SAP or our three categories made no difference for the results. Approximately two-thirds of the sample reported White as their race and another 27% reported Black; approximately 5% was of one of the other race categories.
- Hispanic origin: Approximately 14% of the sample reported being of Hispanic origin. In our analyses, we cross-classified the three race categories we created with the SAP measure of whether a person reported being of Hispanic origin. This mirrors the U.S. Census Bureau's contemporary measures of race and ethnic origin in which race is reported for persons who are not of Hispanic origin. Our Hispanic origin variable was a dummy variable equal to one if a person reported being of Hispanic origin.
- Sex was a dummy variable equal to 1 if the person on probation was a male. More than three-quarters of the sample (77%) was male.
- Education was measured in SAP by respondents reporting the highest level of education completed. We classified these into five categories: Eighth grade or less; ninth through 11th grade; high school graduate; GED; and some college, college graduate, some graduate school. We enter these as separate dummy variables for each category with the eighth grade or less as the omitted variable. One-third of the sample completed high school and another 29% had some college, graduated from college, or attended graduate school.
- To account for probation officers responses to behaviors by persons on probation that may indicate violations or infractions of rules, we measured whether a person had a disciplinary hearing (13.5% of the sample) or were warned about possible rule violations (15%).
- We included the length of sentence imposed (in months) as a control variable.
- Homeownership vs. renting, and contributions towards mortgage or rent.: We measured whether the person owned a home or rented by separate dummy variables

indicating these, and we included a dummy variable that indicated whether the person contributed financially to the mortgage or rent. Approximately one-third (35%) owned a home, while 57% rented. A majority (775) contributed to mortgage or rent payments. Homeownership generally reflects stronger ties to a place or community than does renting. Contributing to a mortgage or rent indicates stronger ties to one's place of residence.

- Marital status was a measured by three dummy variables, one for whether a person was married or widowed, divorced or separated, or never married.
- Finally, we included dummy variables for region of the country where supervised; these region categories were the four major census regions—North, South, Midwest, and West. These region variables also capture elements of the sample design that we use as a variable to cluster standard error estimates. A majority of the sample (62%) resided in the South.

### 3.4. Analytic Techniques

Our key independent variable, missed payments, is a binary variable. We use linear probability models (LPMs) to estimate the relationships between this dependent and its associated independent variables.

We specify our LPM as

$$p = \beta_0 + \beta_1 x_1 + \beta_2 x_2 + \cdots + \beta_k x_k + u$$

where $p = E(Y|X)$, or the expected value (or probability) of an outcome variable $Y$ conditional on values of X. Here, $Y$ equals 1 if a missed payment occurred and zero otherwise. The $\beta_k$ are the coefficients on the independent variables while $\beta_0$ is the intercept of the regression. The $x_k$ represent the independent variables, and $u$ is the error term. We assume that $u$ follows a binomial distribution. We correct for heteroskedastic errors.

Recently, LPMs have gained favor relative to logistic regression models under certain circumstance (Wooldridge 2016; Allison et al. 2020; von Hippel 2015, 2017). The main criticisms of the LPMs are: (1) a binary dependent variable leads to error terms that are non-spherical and therefore violate a key assumption of linear regression; (2) these errors are likely to be heteroskedastic; and (3) predictions from LPMs can fall outside of the 0 to 1 bounds of probabilities. On the other hand, the parameters of an LPM are easier to interpret than those of a logistic regression and use of a linear discriminant function to generate predicated probabilities can overcome the out-of-theoretical bounds of probabilities problem. Specifically, the parameters on a LPM are interpreted as probabilities; if an independent variable is dichotomous, the LPM parameters give the difference in probability between the named and reference group; if an independent variable is continuous, the LPM parameter is interpreted as the change in probability on the dependent variable of a one-unit change in the independent variable.

Logit coefficients, on the other hand, equal the change in the log of the odds of an event (e.g., missed payments) with change in independent variables. Often logit coefficients are converted to odds ratios, but these are frequently mis-interpreted as risk ratios or as changes in probabilities, which odds ratios are not. Predicted probabilities can be generated from logit models and analyses from logits can be presented in terms of predicted probabilities. However, as von Hippel (2015) points out, in most situations the predicted probabilities from a logit give results that are practically indistinguishable from the LPM results. Citing Long (1997), Hippel's point is that for a logit to perform better than an LPM the odds of an event must be a linear function of the independent variables, but the probability must not be. In other words, the relationship between probability and the log of the odds (the logit transformation) must be non-linear. This occurs at the tails of the distribution when the probabilities are, say, above 0.80 or below 0.2. In our application, the mean probability of missing a payment is 0.42, well within the range of acceptability for an LPM.

To address the complex sample design, following Winship and Radbill (1994), we generated unweighted and weighted estimates and compared parameter estimates for

their differences. According to the analyses done by Winship and Radbill if the weighted estimates differ from the unweighted estimates, this may indicate model specification error. The standard errors from the unweighted estimates are efficient. A key understanding whether there is specification error comes from the sample design. If the sample was designed based on the dependent variable—or variables that cause selection on the dependent variable, this can lead to selection bias and specification error. The SAP's sample design was a population-based survey of probationers and not a survey of probationer fees and fines. The weights were calculated sequentially from a base weight (probability of selection based on the sample design) with several non-response adjustments. For details, see United States Department of Justice, Office of Justice Programs, Bureau of Justice Statistics (1997). Because the sample stratification variables could lead to within stratum identification of individuals, the sample design variables were suppressed and are not available. Hence, we could not generate standard errors using the survey design. Our analyses led to a conclusion that the weighted and unweighted estimates were similar, so we had confidence that we minimized the possibility of specification error. We report standard errors based on unweighted estimates.[1]

## 4. Results

### 4.1. Amounts Required and Amounts Paid

Our final sample consists of 2030 persons, 1557 of whom were on active probation and had reported that they had been assessed fees for one or more regular (or other regular) payments. We excluded from the analysis 473 persons who reported that they were not assessed any regular fees. Any contact with the criminal legal system that results in having to go to court carries a fee with along with it. Defendants who are sentenced to probation likely pay the court fee in addition to a potential fine as well as fees associated with their sentence, should they plead guilty. Of the approximately 2 million persons on probation in 1995, 73% were required to make at least one fee/fine payment. By income class, this ranged from two-thirds of the persons in the lowest income classes to 82% in the highest income class (Table 2). The mean amounts ranged from $99 per month in the lowest category to $201 in the highest income class. However, the median amount of monthly payment varied less than the mean amounts. Median monthly payment amounts ranged from $72 to $90. Among income classes, but the difference between the mean and median amounts ranged from approximately $20 to $140, indicating that in the higher income classes, there was more skew in the distribution of monthly payment amounts; Within the highest income classes, several persons had very large monthly fee/fine amounts. The estimated total monthly fee/fine amounts was approximately $170 million.

**Table 2.** Estimated number (weighted counts) of persons on probation required to make payments and weighted estimated amounts of payments, by income class, in current dollars.

| Income Category * | Estimated Number of Persons Required to Make at Least One Regular Payment | Within Income Class, Percent of Persons Required to Make Payments | | Among Those Required to Make at Least One Type of Fee/Fine Payment Estimated Monthly Payment Amounts Ordered | | |
|---|---|---|---|---|---|---|
| | | | | Mean | Median | Total |
| Less than (LT) 9000 | 347,205 | 66.6 | % | $99 | $72 | $34,237,450 |
| 9000 to LT 18,000 | 296,377 | 69.7 | % | 100 | 72 | 29,714,060 |
| 18,000 to LT 27,000 | 296,736 | 76.0 | % | 120 | 90 | 35,754,143 |
| 27,000 to LT 36,000 | 168,391 | 78.4 | % | 145 | 90 | 24,458,134 |
| 36,000 to LT 45,000 | 126,699 | 77.1 | % | 121 | 90 | 15,268,903 |
| 45,000 to LT 90,000 | 178,294 | 81.1 | % | 130 | 76 | 23,260,012 |
| 90,000 or more | 41,048 | 82.4 | % | 201 | 72 | 8,270,476 |
| Total | 1,454,750 | 73.2 | % | 118 | 76 | 170,963,177 |

Source: Authors' analysis of SAP data. Note: * Income classes and amounts in current (2021) dollars.

In the year prior to their SAP interview data, persons on probation paid approximately $765 million in fees and fines in current dollars (or approximately $425 million in 1995 dollars) (Table 3). Collectively, persons in the three lowest income categories paid more than half of the total amount. These three lowest income classes paid smaller mean and median amounts than persons in the highest income classes (e.g., a mean of approximately $440 compared to a mean of over $800 for the persons in the two highest income classes. These differences between mean and total amounts by income class arise from the differences in the numbers of persons in each income class (as shown in Table 2).

**Table 3.** Estimated amounts paid during the past year, by income class.

| Weighted by Final Sample Weights | | | |
|---|---|---|---|
| **Income Category *** | **Estimated Amounts** | | |
| | **Mean** | **Median** | **Total** |
| LT 9000 | $338 | $175 | $111,397,449 |
| 9000 to LT 18,000 | 438 | 240 | 127,711,109 |
| 18,000 to LT 27,000 | 532 | 360 | 144,340,200 |
| 27,000 to LT 36,000 | 800 | 480 | 120,725,318 |
| 36,000 to LT 45,000 | 710 | 480 | 86,470,220 |
| 45,000 to LT 90,000 | 780 | 496 | 135,432,953 |
| 90,000 or more | 954 | 300 | 39,146,392 |
| Total | 555 | 300 | 765,223,641 |

Source: Authors' analysis of SAP data. Note: * Income classes and amounts in current dollars.

### 4.2. Regular Payments and Capacity to Pay

Table 4 presents linear probability model (LPM) estimates of the effect of the number of different types of regular fee/fine types on the probability of missing a payment.[2] The first four models present unweighted results and the fifth shows weighted results, where the weight is the final survey weight for the personal interview sample. In all models, the coefficients are probabilities. When an independent variable is categorical, the coefficients are interpreted as the difference in the estimated probability of missing a payment between the identified category and the reference category. For example, using the results from Model (1), the probability of missing a payment increases by 0.19 points if a person on probation had a disciplinary hearing as compared to a person who did not have a hearing. For continuous variables, the coefficients are interpreted as the change in probability associated with a one-unit change in the variable. For example, using Model (1) results, with each additional regular fee/tine types increase, the probability of missing a payment increases by 0.04. Within each model, the standard errors (S.E.) are reported next to the parameter estimates.

The first four model specifications examine whether the estimates on the key independent variable, the number of types of regular fee/fines, changes with the addition of control variables. These also show the loss of sample occurring with the addition of socio-demographic measures. For example, while the sample loss is comparatively small between Models (2) and (3) when time on probation and sentence length are added, with the addition of past-year income, rent, and contributions to rent, approximately 260 cases are lost as compared to Model (1). However, as the results across Models (1) through (4) show, the loss of sample arising from adding controls does not change the estimates on the number of regular fee/fine types, the key independent variable.

Model (5) presents weighted regression results. Given the complex sample design, this model provides a test of specification error. In reviewing it in comparison to Model (4) (the same regression using unweighted data), there are no indications that the specification in Model (4) had problems. We make this inference by comparing coefficients between the two models. All of the key parameters are within two standard errors of each other; this also holds for control variables. As Winship and Radbill (1994) discuss, when using complex surveys, if weighted regression results differ from the unweighted results, this

may indicate specification error arising from sample design. We find no evidence for this. We use the unweighted results because the standard errors are most efficient.

**Table 4.** Linear probability model estimates of the probability of missing a payment as function of the number of regular fee/fine types ordered. Coefficients are probabilities. Model (5) are weighted estimates using the final weight for the personal interview sample of the SAP.

| Variables | Model (1) | | Model (2) | | Model (3) | | Model (4) | | Model (5): Weighted | |
|---|---|---|---|---|---|---|---|---|---|---|
| | Coef. | S.E. | Coef. | S.E. | Coef. | S.E. | Coef. | S.E. | Coef. | S.E. |
| Number of fee/fine types imposed | 0.0374 *** | (0.0102) | 0.0399 *** | (0.0102) | 0.0432 *** | (0.0105) | 0.0345 *** | (0.0110) | 0.0311 ** | (0.0133) |
| Had a disciplinary hearing | 0.1866 *** | (0.0374) | 0.1733 *** | (0.0372) | 0.1384 *** | (0.0398) | 0.1441 *** | (0.0411) | 0.1685 *** | (0.0501) |
| Was warned about rule violations | 0.1863 *** | (0.0367) | 0.1531 *** | (0.0370) | 0.1687 *** | (0.0383) | 0.1347 *** | (0.0396) | 0.1410 *** | (0.0468) |
| Offense type: | | | | | | | | | | |
| -Rape/sexual assault | 0.0153 | (0.1228) | 0.0757 | (0.1246) | 0.0735 | (0.1269) | −0.0319 | (0.1274) | −0.0808 | (0.1114) |
| -Robbery | 0.1418 | (0.1438) | 0.0877 | (0.1471) | 0.0168 | (0.1514) | −0.1077 | (0.1555) | −0.0663 | (0.1531) |
| -Aggravated assault | 0.0260 | (0.1157) | 0.0017 | (0.1169) | 0.0018 | (0.1194) | −0.0872 | (0.1204) | −0.0943 | (0.1005) |
| -Simple assault | 0.2763 * | (0.1596) | 0.2368 | (0.1594) | 0.2320 | (0.1607) | 0.0898 | (0.1608) | −0.0216 | (0.1587) |
| -Other violent | −0.0478 | (0.2098) | −0.1098 | (0.2085) | −0.1353 | (0.2084) | −0.1307 | (0.2166) | −0.0880 | (0.1577) |
| -Property | 0.1551 | (0.1091) | 0.1292 | (0.1105) | 0.1396 | (0.1129) | 0.0217 | (0.1136) | 0.0136 | (0.0945) |
| -Drug | 0.0859 | (0.1106) | 0.0378 | (0.1120) | 0.0389 | (0.1144) | −0.0677 | (0.1154) | −0.0597 | (0.0963) |
| -Public order | −0.0189 | (0.1111) | −0.0060 | (0.1127) | 0.0071 | (0.1152) | −0.0493 | (0.1158) | −0.0362 | (0.0963) |
| Offense was a felony | −0.0201 | (0.0315) | −0.0164 | (0.0313) | −0.0373 | (0.0335) | −0.0502 | (0.0352) | −0.0329 | (0.0391) |
| Race-Hispanic origin | | | | | | | | | | |
| -White-Hispanic | | | 0.0855 ** | (0.0429) | 0.0918 ** | (0.0438) | 0.0797 * | (0.0455) | 0.0882 * | (0.0512) |
| -Black-non-Hispanic | | | 0.1392 *** | (0.0307) | 0.1639 *** | (0.0318) | 0.1087 *** | (0.0338) | 0.0954 ** | (0.0380) |
| -Black-Hispanic | | | −0.2090 | (0.1582) | −0.1997 | (0.1571) | −0.2948 | (0.1872) | −0.1355 | (0.2114) |
| -Other race-non-Hispanic | | | −0.0785 | (0.0788) | −0.0830 | (0.0793) | −0.1267 | (0.0824) | −0.1587 *** | (0.0610) |
| -Other race-Hispanic | | | 0.0584 | (0.0781) | 0.0407 | (0.0817) | 0.0384 | (0.0837) | 0.0604 | (0.0872) |
| Person was a male | | | −0.0437 | (0.0305) | −0.0452 | (0.0313) | 0.0128 | (0.0340) | −0.0178 | (0.0390) |
| Age in years | | | −0.0047 *** | (0.0012) | −0.0056 *** | (0.0013) | −0.0041 *** | (0.0014) | −0.0046 *** | (0.0014) |
| Time on probation (in months) | | | | | 0.0052 *** | (0.0016) | 0.0059 *** | (0.0016) | 0.0069 *** | (0.0015) |
| Time on probation squared | | | | | −0.0000 ** | (0.0000) | −0.0000 ** | (0.0000) | −0.0000 *** | (0.0000) |
| Length of sentence imposed | | | | | −0.0000 | (0.0010) | 0.0003 | (0.0011) | 0.0002 | (0.0012) |
| Past-year personal income ^ | | | | | | | | | | |
| -$9 K–LT $18 K | | | | | | | −0.0889 ** | (0.0408) | −0.0909 * | (0.0502) |
| -$18 K–LT $27 K | | | | | | | −0.1547 *** | (0.0415) | −0.1548 *** | (0.0495) |
| -$27 K–LT $36 K | | | | | | | −0.2339 *** | (0.0509) | −0.2447 *** | (0.0553) |
| -$36 K–LT $45 K | | | | | | | −0.2405 *** | (0.0556) | −0.2259 *** | (0.0608) |
| -$45 K–LT $90 K | | | | | | | −0.3229 *** | (0.0525) | −0.3178 *** | (0.0540) |
| -$90 K or more | | | | | | | −0.2690 *** | (0.0879) | −0.2633 *** | (0.0830) |
| Rented a place | | | | | | | 0.0930 *** | (0.0304) | 0.0676 ** | (0.0337) |
| Contributed to rent | | | | | | | −0.0299 | (0.0370) | 0.0033 | (0.0419) |
| Region | | | | | | | | | | |
| -Midwest | | | | | | | 0.0402 | (0.0575) | 0.0596 | (0.0650) |
| -South | | | | | | | −0.0447 | (0.0492) | −0.0644 | (0.0572) |
| -West | | | | | | | −0.0013 | (0.0589) | −0.0066 | (0.0699) |
| Constant | 0.2252 ** | (0.1129) | 0.3795 *** | (0.1239) | 0.3304 ** | (0.1283) | 0.4902 *** | (0.1405) | 0.5081 *** | (0.1367) |
| Observations | 1414 | | 1400 | | 1304 | | 1158 | | 1158 | |
| R-squared | 0.0697 | | 0.0978 | | 0.1175 | | 0.1767 | | 0.1867 | |

Standard errors in parentheses. *** $p < 0.01$, ** $p < 0.05$, * $p < 0.1$. Source: Authors' analysis of the SAP data. ^ Past-year total personal income categories are shown in current dollars; all analyses done using 1995 dollars.

The main results from Table 4 are as follows: An increase in the number of fee/fine types results in an increase in the probability of missing a payment. While a unit increase in the number of types result in the 0.03 change in the probability of missing a payment

(on a base probability of 0.42), for persons with eight types as opposed to 1 or 2 types of fee/fine orders, the effect of the number of fee/fine types can be as high as 0.21. These results are consistent with our hypotheses.

Turning to other variables, the effects of disciplinary hearings and being warned of rule violations have comparatively large effects, increasing the probability of missing a payment by approximately 0.14 (in Model 4). We found no statistically significant effects of the type of offense or whether a person was supervised for a felony offense.

We introduced race and Hispanic origin by cross-classifying these two variables to differentiate persons who identified both a race group and Hispanic origin from those who did not. Using Model (4) results, the reference group, or omitted category from the regressions is White-non-Hispanics; by comparison White-Hispanics and Black-non-Hispanics were more likely to miss a payment (approximately 7 and 11 percentage points, respectively). Males were no more likely than females to miss payments, and older persons were less likely than younger persons to miss payments. Time on probation was non-linearly related to missing payments; as time on probation increased, the probability of missing a payment increased up until approximately 73 months, and then it decreased. (Mean time on probation until the SAP interview was approximately 20 months.) As the length of time on probation is a function of both the length of sentence imposed and behavior, as persons who have infractions may have their time on active probation extended, as compared to those who may be discharged to inactive status if they have no infractions, the negative coefficient on the squared term on time on probation may indicate that these persons had prior behaviors that extended their terms and that eventually their likelihood of missing a payment fell.

Personal income was negatively correlated with missing payments. As expected, as income increased, the probability of missing a payment fell. The two highest income classes, were between 33 and 27 percentage points less likely to miss payments than the lowest income class. Renters were more likely to miss payments than home owners, and there were no statistically significant differences among the regions of the country in the probability of missing payments.

In terms of capacity to pay, probationers who received welfare benefits were approximately 13 percentage points more likely to miss a payment, when all controls are added (model 4). Conversely, as past-year income increased, the probability of missing a payment fell. Probationers in the lowest income categories are predicted to miss payments with a probability exceeding 60%, while those in the highest past-year income categories had a 20% of missing a payment. Of note is that the predicted probability of missing a payment increases throughout the first six years on probation and then declines. If missing payments extends probation terms, this relationship could be endogenous, so we do not want to imply a causal effect of time on probation and missed payments.

*4.3. Missing Restitution Payments*

Next, we turn attention to the question of whether the types of fees/fines affect missed payments. Consistent with the background and theory we presented, we argue that fee types associated with maintaining and supporting criminal justice system agencies would lessen the probability of missing a payment when compared to the effects of being ordered to pay restitution. These types of fees include probation service fees, court costs, treatment fees, and others that support justice agency operations. We speculate that the priority of paying these financial supports would be communicated to persons on probation, and that paying them would take precedence of paying restitution, which does not go to support agency costs.

In Table 5, we present the results of linear probability models of the probability of missing a payment conditional on the presence of specific types of fees. Table 5 presents results from five regression models; models (1) through (4) show unweighted regression results and model (5) shows weighted results, where the weight (as before) is the final survey weight for the personal interview SAP data. Coefficients are estimated probabilities.[3] As

with Table 4, we focus on Model (4) results, our final model, given that it and the weighted estimates in Model (5) are comparable and do not indicate specification error.

**Table 5.** Linear probability model (LPM) estimates of the probability of missing a payment as a function of the specific types of fees/fines imposed.

| Variables | Model (1) | | Model (2) | | Model (3) | | Model (4) | | Model (5) Weighted | |
|---|---|---|---|---|---|---|---|---|---|---|
| | Coef. | S.E. | Coef. | S.E. | Coef. | S.E. | Coef. | S.E. | Coef. | S.E. |
| Restitution ordered | 0.1370 *** | (0.0307) | 0.0944 *** | (0.0361) | 0.1070 *** | (0.0386) | 0.0975 ** | (0.0401) | 0.0831 * | (0.0475) |
| Court costs ordered | 0.0494 | (0.0303) | 0.0597 * | (0.0321) | 0.0620 * | (0.0334) | 0.0563 * | (0.0302) | 0.0428 | (0.0336) |
| Probation fee ordered | 0.0252 | (0.0292) | 0.0688 ** | (0.0301) | 0.0656 ** | (0.0315) | 0.0581 * | (0.0326) | 0.0442 | (0.0367) |
| Fine ordered | 0.0327 | (0.0302) | 0.0268 | (0.0318) | 0.0315 | (0.0330) | | | | |
| Drug treatment fee | 0.0396 | (0.0467) | −0.0038 | (0.0501) | 0.0426 | (0.0518) | | | | |
| Community service fee | −0.0598 | (0.0592) | −0.0502 | (0.0611) | −0.0123 | (0.0620) | | | | |
| Lump-sum payment | | | −0.0602* | (0.0308) | −0.0607 * | (0.0315) | −0.0441 | (0.0324) | −0.0473 | (0.0348) |
| Monthly payment amount | | | | | −0.0003 * | (0.0002) | 0.0000 | (0.0002) | 0.0001 | (0.0003) |
| Had a disciplinary hearing | | | 0.1853 *** | (0.0375) | 0.1409 *** | (0.0403) | 0.1457 *** | (0.0420) | 0.1563 *** | (0.0516) |
| Was warned about rule violations | | | 0.1856 *** | (0.0367) | 0.1689 *** | (0.0389) | 0.1431 *** | (0.0403) | 0.1497 *** | (0.0486) |
| Offense type: | | | | | | | | | | |
| -Rape/sexual assault | | | 0.0454 | (0.1230) | 0.0907 | (0.1273) | −0.0337 | (0.1276) | −0.0623 | (0.1223) |
| -Robbery | | | 0.1563 | (0.1437) | 0.0356 | (0.1511) | −0.0936 | (0.1548) | −0.0348 | (0.1639) |
| -Aggravated assault | | | 0.0424 | (0.1158) | 0.0162 | (0.1194) | −0.0670 | (0.1201) | −0.0624 | (0.1123) |
| -Simple assault | | | 0.2983 * | (0.1600) | 0.2169 | (0.1631) | 0.1018 | (0.1618) | −0.0012 | (0.1565) |
| -Other violent | | | −0.0566 | (0.2096) | −0.1440 | (0.2078) | −0.1512 | (0.2162) | −0.0986 | (0.1818) |
| -Property | | | 0.1518 | (0.1099) | 0.1334 | (0.1134) | 0.0094 | (0.1138) | 0.0184 | (0.1080) |
| -Drug | | | 0.1159 | (0.1110) | 0.0674 | (0.1150) | −0.0337 | (0.1155) | −0.0189 | (0.1092) |
| -Public order | | | 0.0098 | (0.1114) | 0.0272 | (0.1154) | −0.0196 | (0.1156) | 0.0070 | (0.1086) |
| Offense was a felony | | | −0.0347 | (0.0322) | −0.0522 | (0.0345) | −0.0631 * | (0.0358) | −0.0453 | (0.0408) |
| Time on probation (in months) | | | | | 0.0049 *** | (0.0016) | 0.0060 *** | (0.0016) | 0.0070 *** | (0.0015) |
| Time on probation squared | | | | | −0.0000 ** | (0.0000) | −0.0000 *** | (0.0000) | −0.0001 *** | (0.0000) |
| Sentence imposed (in months) | | | | | −0.0001 | (0.0011) | 0.0003 | (0.0011) | 0.0003 | (0.0013) |
| Past-year personal income: ^ | | | | | | | | | | |
| -$9 K–LT $18 K | | | | | | | −0.0495 | (0.0421) | −0.0714 | (0.0527) |
| -$18 K–LT $27 K | | | | | | | −0.1218 *** | (0.0428) | −0.1256 ** | (0.0515) |
| -$27 K–LT $36 K | | | | | | | −0.1944 *** | (0.0527) | −0.2180 *** | (0.0588) |
| -$36 K–LT $45 K | | | | | | | −0.1860 *** | (0.0574) | −0.1812 *** | (0.0645) |
| -$45 K–LT $90 K | | | | | | | −0.2568 *** | (0.0554) | −0.2689 *** | (0.0590) |
| -$90 K or more | | | | | | | −0.1894 ** | (0.0907) | −0.2104 ** | (0.0898) |
| Race-Hispanic origin | | | | | | | | | | |
| -White-Hispanic | | | | | 0.0831 * | (0.0441) | 0.0809 * | (0.0459) | 0.0901 * | (0.0514) |
| -Black-non-Hispanic | | | | | 0.1459 *** | (0.0326) | 0.0925 *** | (0.0343) | 0.0813 ** | (0.0394) |
| -Black-Hispanic | | | | | −0.2017 | (0.1569) | −0.2472 | (0.2045) | −0.1093 | (0.2341) |
| -Other-non-Hispanic | | | | | −0.0950 | (0.0791) | −0.1151 | (0.0819) | −0.1664 *** | (0.0608) |
| -Other-Hispanic | | | | | 0.0425 | (0.0816) | 0.0694 | (0.0872) | 0.0852 | (0.0884) |
| Person was a male | | | | | −0.0447 | (0.0318) | 0.0263 | (0.0356) | −0.0063 | (0.0408) |
| Age in years | | | | | −0.0056 *** | (0.0013) | −0.0036 ** | (0.0014) | −0.0044 *** | (0.0015) |
| Received public assistance | | | | | | | 0.1139 *** | (0.0434) | 0.1032 * | (0.0544) |
| Education level completed: | | | | | | | | | | |
| −9th thru 11th grade | | | | | | | 0.1787 *** | (0.0680) | 0.1555 ** | (0.0779) |
| -High school | | | | | | | 0.1571 ** | (0.0640) | 0.0941 | (0.0734) |
| -GED | | | | | | | 0.1300 * | (0.0721) | 0.1201 | (0.0831) |
| -Some college, college, grad.sch. | | | | | | | 0.0840 | (0.0647) | 0.0808 | (0.0727) |

**Table 5.** *Cont.*

| Variables | Model (1) | | Model (2) | | Model (3) | | Model (4) | | Model (5) Weighted | |
|---|---|---|---|---|---|---|---|---|---|---|
| | Coef. | S.E. | Coef. | S.E. | Coef. | S.E. | Coef. | S.E. | Coef. | S.E. |
| Rented | | | | | | | 0.0865 *** | (0.0309) | 0.0618 * | (0.0341) |
| Contributed to rent | | | | | | | −0.0069 | (0.0372) | 0.0305 | (0.0412) |
| Constant | 0.3363 *** | (0.0305) | 0.2059 * | (0.1162) | 0.3486 *** | (0.1321) | 0.2385 | (0.1516) | 0.2895 * | (0.1563) |
| Observations | 1557 | | 1414 | | 1279 | | 1116 | | 1116 | |
| R-squared | 0.0198 | | 0.0782 | | 0.1284 | | 0.1910 | | 0.1931 | |

Standard errors in parentheses. *** $p < 0.01$, ** $p < 0.05$, * $p < 0.1$. Source: Authors' analysis of SAP data. ˆ Income classes reported in current dollars; analyses based upon 1995 dollars.

We find first that the effects of restitution are consistent across specifications, and that having a restitution order increases the probability of missing a payment by approximately 10 percentage points, as show in Model (4). The magnitude of the effect of restitution is almost twice as large as the effect probation and court cost fees. This result is consistent with our hypotheses about restitution.

In Model (4) disciplinary hearings and rule violation warnings also had large effects on missing payments; each increased the probability of missing a payment by approximately 14 percentage points. Including disciplinary hearings and warnings of rule violations in the regressions caused the effects of probation and court cost fees to become significant. This is indicated by comparing the parameter estimates between Models (1) and (2). The effect of including these variables was largest for probation fees, where the parameter estimate doubled between Models (1) and (2) (from an insignificant 2.5 percentage point difference to approximately a 7 percentage point and statistically significant difference) when disciplinary hearings and rule violation warnings were added; the magnitude of the probation fee effect remained at that level across the remaining Models (3) and (4). In order for including disciplinary hearings and rule violations in the regressions to lead to an increase in the effect of probation and court cost fees on missed payments, it must be the case that hearings and warnings take away a portion of the variation between these two fee types and missed payments that contributed to their high variance. Substantively, this may mean that persons with probation or court cost fees are more likely to miss payments given that disciplinary hearings and rule violation warnings are held constant; or, once the regression is purged of the effects of hearings and warnings, people with probation and court fees are more likely to miss payments. Persons who do not have hearings or warnings are likely to engage in fewer behaviors that would result in the hearings, so this poses the question about why the "better-behaved" persons would be more likely to miss payments. One explanation is that they consider the fees to be onerous and choose not to pay them. Under this theory, the imposition of these fees lessens the legitimacy of the justice system in the eyes of the person ordered to pay the fees, leading to an increase in the probability of missing a payment.

In Model (4), offense type and whether the offense was a felony were not statistically significant, indicating the type of offense of conviction had no bearing on whether a person on probation missed a payment. Similarly, the length of sentence imposed was not significant. As with the result in Table 4, the length of time on probation is non-linearly related to missing payments, first increasing and then decreasing as the length of time on probation increases.

Past-year personal income is, as we hypothesized, negatively correlated with missing payments. The probability of missing a payment decreases in moving from the lower income levels, but above the income level of $36 K to $45 K, it does not decrease further. Still persons in these higher income levels are approximately 20 percentage points less likely to miss a payment than persons in the lowest income category. Receiving public assistance also increases the probability of missing payments. In short, low income persons on public assistance are most likely to miss payments.

We investigated the cross between race and Hispanic origin and found that White-Hispanics and Black-non-Hispanics were more likely than White-non-Hispanics to miss payment (approximately 9 percentage points more likely).

We note that across specifications (2), (3), and (4), there is a loss of sample that arises primarily from including socio-demographic measures. However, including these variables in the regressions does not materially change the effects of the key variables, the types of fees.

## 5. Discussion

Our results support our hypotheses that probationers with the least amount of income will have the most trouble paying their legal debt. Indeed, people on probation whose past-year income fell in the lowest categories had a significantly higher probability of missing a payment than those in the upper income categories (60% vs. 20%). In addition to disparities in the likelihood of missing a payment based on income, we found that as the number of regular payments increased so did the likelihood of missing a payment (e.g., paying toward a fine versus paying toward a fine, court fee, and restitution). These findings are somewhat unsurprising considering that contemporary literature on fines and fees tends to focus on the impact of LFOS on economically marginalized populations. On the other hand, that restitution was found to be the sole legal financial obligation associated with missing a payment is intriguing and perhaps our most important finding. We hypothesized that revenue generating LFOs such as fees would take precedence over other types such as fines and restitution. Our reasoning for this hypothesis, as stated in the Current Study section, is that people on probation may perceive the consequences of missing certain LFOs differently. For instance, if a judge or probation officer encourages them to pay their supervision fees or fines first, that might be prioritized over restitution out of fear of receiving a technical violation or revocation. Alternatively, the collection of fees and fines could be better managed than the collection of restitution, discussed below. Regardless, future research might focus on how perceptions of certain LFOs impacts payment practices.

Our finding on restitution is additionally troubling within the context of research and policy efforts since 1995. First, the victim's rights movement was growing rapidly during the 1990s, and restitution payments were at the forefront of victim's advocates' discussions. According to a 1998 report from the Office for Victims of Crime, 29 states had mandatory restitution laws by 1995, up from 8 in 1982 (United States Department of Justice, Office of Justice Programs, Office for Victims of Crime 1998). The following year, congress enacted the Mandatory Restitution Act of 1996 which required restitution in all federal cases involving property, bodily injury, loss of life, and loss of income. Further, in the years 1995 and 1996, the OVC funded program development for standards on state compensation programs. The goal was to set guidelines for "effective outreach, training, and communication; expeditious and accurate claims processing; good decision making; and sound financial planning" (United States Department of Justice, Office of Justice Programs, Office for Victims of Crime 1998, p. 326). Although these efforts, among others, have been made to increase restitution payments, the rate of collection remains abysmal to this day. As previously mentioned, the U.S. Government Accountability Office estimated that federal judges assessed 33.9 BN in restitution from 2014 to 2016. Only nine percent of that number was collected, amounting to 2.95 BN (United States Government Accountability Office 2018). The GAO cited peoples' inability to pay as one of the leading issues related to collection of restitution. Therefore, there is evidence to support the claim that the problem persists at the federal level despite the efforts to increase restitution payments.

Second, there are themes in the literature on LFOs that highlight the importance of LFOs containing some philosophical purpose, whether it be retribution or restoration. In their interviews with people who had legal debt, Pattillo and Kirk (2020, p. 61) found they "believed in retributive justice. They recognized that there are consequences to law breaking

and accepted the authority of the judicial system to impose penalties. Their statements of recognition were robust and included elements of contrition and support for deterrence and rehabilitation." Thus, if revenue generating LFOs (e.g., court costs and supervision fees) are being prioritized over payments that hold philosophical value (restitution) people may be less likely to accept the authority of the judicial system.

Research does indeed support the claim that restitution has rehabilitative and restorative effects for both the victim and offender. In their experiment testing the impact of information and rationale on making restitution payments, Ruback et al. (2018) found that non-recidivists were more likely to have made monthly payments, and paid more toward their total debt, than recidivists. From the victim's standpoint, restitution has also been shown to have rehabilitative and restorative effects and has also been linked to themes related to legitimacy such as future willingness to cooperate with the courts (Ruback et al. 2018). Certainly, missing restitution payments may harm the legitimacy of the courts. Victims who do not receive restitution payments may believe that their experience was not taken seriously and could be less likely to cooperate in the future. Alternatively, if they believe they are heard by the court and if the convicted person makes direct payments to them, legitimacy could be enhanced.

Future research on legal financial obligations should focus on the relationship between philosophical purpose and legitimacy. Much of the research on legitimacy in our justice system has focused on the front end; namely, police contact. We propose that it is just as important to examine whether legitimacy is maintained throughout the entire justice system, from initial police contact through corrections. Legitimacy bolsters a sense of obedience and willful compliance to an authority such as the government. Research has shown that when government officials act in a procedurally fair manner (e.g., neutrality, respectfulness, and willingness to listen) and are viewed by the public as trustworthy, legitimacy is enhanced (Levi et al. 2009). Inequalities in punishment severity based on wealth, conflicts of interest benefitting the government, and mistranslations from policies to practice may all contribute to a legitimacy problem for the courts and community supervision.

We additionally want to emphasize the importance of another national data collection effort on LFOs, such as the SAP. Countless researchers studying LFOs have offered this recommendation before us. We join them in these calls. Until we are able to study LFOs using nationally representative data, recommendations and findings may only be applicable at the local level.

*Limitations*

Apart from the date of the data and the need for more recent and timely surveys of probationers, the use of fees and fines varies considerably among jurisdictions, and we are unable to capture how this variation might explain probationers missing payments. The best we could do on this was to include the region of the probationer, and while that showed that probationers in the South were less likely to miss payments, we also know that among states within the South there were differences in fee/fine use. Because of the local nature of supervision, jurisdiction-specific effects in the imposition and missing of payments can help understand the extent to which fee- and non-fee funding mechanisms contribute to LFO burden and missed payments.

As we pointed out, the SAP excludes probationers of interest, such as absconders and persons in jail or prison. The latter two groups are particularly important in light of current arguments about how fees and fines contribute to the size of both probation and custodial populations. The BJS surveys of inmates could be a source of this information, but these surveys already capture lots of information about many domains, and adding additional questions about fees and fines as a cause of incarceration would likely add burden with potentially little payoff. Rather, the connection between missing payments and custody might better be addressed through linking administrative data from both areas.

Our measures of debt burden relative to ability to pay need to be improved. For example, conditioning income on family structure and other budget items would improve the measures of ability to pay, the SAP data on these are missing to a large degree. Future surveys of probationers could consider linking survey results with administrative data such as unemployment insurance data. The effects of missed payments on revocations, extending the length of probation, or other supervision outcomes could not be determined. Improved measures of these would help in understanding the consequences of missing payments. Here, again, linking survey to probationer administrative data at the jurisdiction level would be useful. Finally, while the national-level data give a sense of the overall effects of fees on missed payments, future research could overcome the limitations of national-level results by comparing differences among jurisdictions. This may require probation departments to collect better data on fees and fines than seems to exist (Harris et al. 2016), and doing so would contribute both to improving operations and to research.

**Author Contributions:** Conceptualization, M.L.W.; methodology, W.J.S.; software, W.J.S.; validation, M.L.W. and W.J.S.; formal analysis, W.J.S.; investigation, M.L.W. and W.J.S.; resources, M.L.W. and W.J.S.; data curation, M.L.W.; writing—original draft preparation, M.L.W.; writing—review and editing, M.L.W. and W.J.S.; visualization, W.J.S.; supervision, W.J.S.; project administration, M.L.W. All authors have read and agreed to the published version of the manuscript.

**Funding:** This research received no external funding.

**Institutional Review Board Statement:** Not applicable.

**Informed Consent Statement:** Not applicable.

**Data Availability Statement:** The data presented in this study are openly available in the Inter-University Consortium for Political and Social Research at https://doi.org/10.3886/ICPSR02039.v1 (accessed on 10 November 2021), reference number 2039.

**Acknowledgments:** We wish to thank the two anonymous reviewers for their comments on a prior version of this paper.

**Conflicts of Interest:** The authors declare no conflict of interest.

## Notes

[1]    Weighted estimates are available from the authors upon request.

[2]    We have also estimated these same regressions using a logit specification and can make these available upon request. Given debates about LPM and logits regarding predicted probabilities, we examined whether there were any differences in estimated effects between the logit and LPM; we found none. We compared predicted probabilities from the LPM and logit and found small and insignificant differences. We attribute this to the fact that the baseline probability of missing payments is approximately 42%, and as von Hippel (2017) has pointed out, when the underlying probabilities are of that magnitude, the logit and LPM estimates yield very similar results.

[3]    As with Table 4, for the models in Table 5, we also estimated logit models to assess whether the functional form affected the estimates and their statistical significance. The logit and LPM models yield comparable probabilities and tests of significance. The logit results are available upon request of the authors.

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
