# Peer review of "Legal Financial Obligations and Probation: Findings from the 1995 Survey of Adults on Probation"

_socsci, doi:10.3390/socsci10120450_

Round 1
Reviewer 1 Report
The current study provides statistical analysis of experiences and outcomes related to legal financial obligations for probationers. The study is unique in that is uses a representative dataset of probationers and is able to capture an array of important factors in determining outcomes related to LFOs. In present form, the study requires some additional clarity and information to attend to concerns about the data and interpretations.
The authors present a very thorough, yet succinct, review of the LFO literature, however there are a few points for additional information or explanation. First, the authors indicate individuals may be charged “third-party supervision fees” (p. 2/line 93)—can the authors explain this a bit more. Similarly, there are references to probation fees, but not much explanation. It seems relevant that probation fees (which were not as common in 1995) be discussed clearly.
When discussing the philosophical purposes of LFOs the authors allude to legitimacy of the LFOs, then provide some more detail, but not much, in the Current Study section. The conversation about legitimacy is useful as a byproduct of incurring and paying fees (both in seeing the fees as legitimate, but also in tandem with the legitimacy of the justice system overall). Since the authors are not assessing perceived legitimacy, it may not be necessary to include this in the Current Study section. If the authors are using legitimacy frameworks to hypothesize why someone might miss certain types of payments, much more attention to legitimacy needs to be paid in the literature review.
It seems prudent to provide some context for the reader for using the 1995 SAP data earlier in the manuscript. When reading the abstract, I was immediately cautious of using 25 year old data related to financial issues among probationers since so much has evolved in the area of LFOs since the survey, but the authors provide appropriate rationale in Section 3.1. Offering just a bit of context for accepting this analysis as useful to present day in the Current Study section will set up the rest of the study successfully.
In H2 the authors reference procedural fairness, but have not provided sufficient background to support why this concept is theoretically tied to paying restitution. I also question the rationale for hypothesizing that individuals will be more likely to pay system-user fees (compared to restitution) because they keep the system operating. Legitimacy and procedural justice would suggest the individuals need to see these fines/fees as fair and legitimate in order to comply, but do they view them that way? Much research supports the idea that individuals are more likely to pay things like restitution because they understand where the money is going or feel guilt about the harm they caused; is it the case that perhaps there is more pressure to pay user fees by the system? More explanation is needed here.
The information provided about the prior SAP study (p.6/line 257-264) needs more explanation to show the contrasts to the current study, or may not need to be included at all. If retained the authors need to explain further the refer to social location (wealth) as this is not in the front end as clearly. I am not sure it is needed other than saying you’re expanding on prior work by considering ability to pay.
When describing the BJS dataset and collection, can the authors speak a bit more to how the data were collected? There is mention of “personal interviews” but what does this mean? Is there only survey data or is there also qualitative data? Do the authors have a refusal rate? Can the authors say more about representativeness of the interview sample?
A bit more explanation is needed related to the measures. When the authors say one of the independent variables is “number of regularly fines/fees imposed” are they referring to the count of each type of fee the individuals are paying? What is included in the “other fee types”? Can the authors describe/define “inability to pay”? They mention a dummy variable for each income class, but what does this refer to and how is it included in the analysis?
The sample size reduces from 2,030 to 1,463—can the authors provide information on the missing cases? There are also sample differences in the models but no explanation as to why or what type of missing data analysis was conducted.
When providing the 1995 average income, can the authors provide today’s equivalent income for context?
The sentences before introducing the Tables (p.8 /lines 392-395) seem out of place. I am not sure they are needed.
The Tables could be more explanatory and formatted for the reader (The tables appear copied from Excel and not formatted for the reader). In Table 1, what is the percentage indicating? In Tables 1 and 2, what does GE stand for? What does missing refer to? In Table N (is this Table 3) the authors say N of regular payments—does this mean the number of fines and fees regularly imposed? There are also several indicators not included in the models, but I am not sure why? Is it because they are not significant? Why not include them, especially since past year income is relevant as an indicator of ability to pay? Table 4 (which is labeled Table 2) has the variables labels from the statistical software which the reader can intuit, but would be better presented in a reader-friendly fashion. I would encourage the inclusion of all variables.
Finally, I would encourage the authors to reduce/eliminate the use of the term “offender” and align the terminology with current standards in the APA about using person-first language.
Author Response
We appreciate your insight on our paper. Attached is our response which also contains our response to reviewer 2.

Reviewer 2 Report
This is an interesting article that examines type of fees, fines and restitution on whether individuals miss a payment. Some of the conclusions are logical (those with less money have are more likely to miss a payment). The finding about restitution is very interesting.
As noted by the authors, this is an old data set (1995) with fines, fees, restitution increasing in payments. Nonetheless, the authors have posed interesting questions that are important.
- Background literature--The literature review does not include the more recent literature on how the amount of fines/fees/restitution affects probation outcomes. Nor does it explore how different types of fines/fees/restitution have evolved over time.
- Methods
- it should be noted that the data set does not include individuals on federal probation
- It should be noted what type of outcomes are available for individuals such as rearrest, revocation, return to incarceration. If these outcomes are available then perhaps a model could be provided that uses missed payments and types of LFO to predict outcomes from supervision.
- The tables are not clear. What are the theoretical basis for each model.
- Tables 1 and 2 can be combined. It is currently confusing and the labelling is unclear.
- Table 3 is poorly labeled and it is unclear whether they are showing OR or beta coefficients. The control variables in models 2-4 are unclear why they were selected. Having predictor values of race/ethnicity (white, hispanic, white-hispanic) in a model would appear that there is a problem of multicollinearity. Also in each model there is a loss of sample (model 1 has around 1500 while model 4 has about 1000) with little discussion of why cases are lost, and implications of the lost cases (are they different than what is model 1).
- The models do not include offense type--drug, property, personal or other. why?
- The same comments are in Table 4 regarding losing cases, race/ethnicity variables, etc.
- It is unclear why sentence length is not a variable in the equation since length of time on probation might affect payment.
- The findings about restitution are interesting but the explanation does not make sense. More on restitution and why this might be the one that a person does not pay (how is it managed, etc.). Why is this considered procedural fairness? does the restitution variable vary by type of offense? other required payment?
- The study does not include variables other than region regarding how the funds are used. In Texas, fees are used to pay for the probation agency which is unique. Is there differences in regional use of fees.
- The study does not provide a good context for understanding why this is an important study--the limitations do not include information about interviews with individuals regarding the impact of LFO on their probation experiences. This is a missing issue.
Author Response
We appreciate your feedback on our paper. Attached is our response which also contains our responses to reviewer 1.

Round 2
Reviewer 1 Report
The revised article addresses many of the concerns and critiques related to the original presentation. There are tables in text that are unclear if they are supposed to be there (Table N and Table 2) or are replaced by Tables 4 and 5. There are two other sentences that need clarification:
The sentence on Pg 6 274-275 needs a bit of clarity, or perhaps its just a word or two missing.
The information on p. 7 lines 309-310 needs some clarification. What are self-representing agencies and how does this contrast to those selected systematically and based on what system.
Reviewer 2 Report
This is a much improved manuscript. I appreciate the attention to the details. I have 3 questions:
On Table 1 you report that 65% of the offenses are felonies but the majority of the offenses are public order offenses. This does not make sense.
Beginning on pg 15, there are a number of tables that are not referenced and appear to be duplicative. I can not figure out which ones you desire to keep or not.
Maybe you can add a sentence why data from 1995 is relevant to review for.
